# Clinical Impact of Enteral Protein Nutritional Therapy on Patients with Obesity Scheduled for Bariatric Surgery: A Focus on Safety, Efficacy, and Pathophysiological Changes

**DOI:** 10.3390/nu15061492

**Published:** 2023-03-20

**Authors:** Giuseppe Castaldo, Luigi Schiavo, Imma Pagano, Paola Molettieri, Aurelio Conte, Gerardo Sarno, Vincenzo Pilone, Luca Rastrelli

**Affiliations:** 1NutriKeto_LAB UNISA-“San Giuseppe Moscati” National Hospital (AORN), Contrada Amoretta, 83100 Avellino, Italy; 2Department of Medicine, Surgery and Dentistry “Scuola Medica Salernitana”, University of Salerno, 84081 Baronissi, Italy; 3NBFC, National Biodiversity Future Center, 90133 Palermo, Italy; 4Department of Pharmacy, University of Salerno, Via Giovanni Paolo II 132, Fisciano, 84084 Salerno, Italy; 5General Surgery and Kidney Transplantation Unit, San Giovanni di Dio e Ruggi d’Aragona University Hospital, 84131 Salerno, Italy

**Keywords:** weight loss, enteral protein nutritional therapy, obesity surgery, preoperative care

## Abstract

Background: Ketogenic diet-induced weight loss before bariatric surgery (BS) has beneficial effects on the reduction in the liver volume, metabolic profile, and intra- and post-operative complications. However, these beneficial effects can be limited by poor dietary adherence. A potential solution in patients showing a poor adherence in following the prescribed diet could be represented by enteral nutrition strategies. To date, no studies describe the protocol to use for the efficacy and the safety of pre-operative enteral ketogenic nutrition-based dietary protocols in terms of weight reduction, metabolic efficacy, and safety in patients with obesity scheduled for BS. Aims and scope: To assess the clinical impact, efficacy, and safety of ketogenic nutrition enteral protein (NEP) vs. nutritional enteral hypocaloric (NEI) protocols on patients with obesity candidate to BS. Patients and methods: 31 NEP were compared to 29 NEI patients through a 1:1 randomization. The body weight (BW), body mass index (BMI), waist circumference (WC), hip circumference (HC), and neck circumference (NC) were assessed at the baseline and at the 4-week follow-up. Furthermore, clinical parameters were assessed by blood tests, and patients were asked daily to report any side effects, using a self-administered questionnaire. Results: Compared to the baseline, the BW, BMI, WC, HC, and NC were significantly reduced in both groups studied (*p* < 0.001). However, we did not find any significative difference between the NEP and NEI groups in terms of weight loss (*p* = 0.559), BMI (*p*= 0.383), WC (*p* = 0.779), and HC (*p* = 0.559), while a statistically significant difference was found in terms of the NC (NEP, −7.1% vs. NEI, −4%, *p* = 0.011). Furthermore, we found a significant amelioration of the general clinical status in both groups. However, a statistically significant difference was found in terms of glycemia (NEP, −16% vs. NEI, −8.5%, *p* < 0.001), insulin (NEP, −49.6% vs. NEI, −17.8%, *p* < 0.0028), HOMA index (NEP, −57.7% vs. NEI, −24.9%, *p* < 0.001), total cholesterol (NEP, −24.3% vs. NEI, −2.8%, *p* < 0.001), low-density lipoprotein (NEP, −30.9% vs. NEI, 1.96%, *p* < 0.001), apolipoprotein A1 (NEP, −24.2% vs. NEI, −7%, *p* < 0.001), and apolipoprotein B (NEP, −23.1% vs. NEI, −2.3%, *p* < 0.001), whereas we did not find any significative difference between the NEP and NEI groups in terms of aortomesenteric fat thickness (*p* = 0.332), triglyceride levels (*p* = 0.534), degree of steatosis (*p* = 0.616), and left hepatic lobe volume (*p* = 0.264). Furthermore, the NEP and NEI treatments were well tolerated, and no major side effects were registered. Conclusions: Enteral feeding is an effective and safe treatment before BS, with NEP leading to better clinical results than NEI on the glycemic and lipid profiles. Further and larger randomized clinical trials are needed to confirm these preliminary data.

## 1. Introduction

In patients with morbid obesity scheduled for bariatric surgery (BS), pre-operative moderate weight loss (~10%) and liver volume and steatosis reduction are desirable [1,2,3,4,5]. Liver steatosis in patients suffering from morbid obesity undergoing BS increases the liver volume and may complicate the surgical procedure when the liver’s left lateral section is massively enlarged, limiting the access to the esophagogastric junction and increasing the risk of laceration of the soft fatty liver with consequent bleeding [6,7]. In turn, these difficulties may result in an increased operative time, suboptimal surgery, and an increased rate of conversion to open surgery [7].

With the aim to obtain moderate weight loss and liver volume and steatosis reduction before BS, several dietary protocols have been introduced over time, among them very low-calorie diets (VLCDs) and very low-calorie ketogenic diets (VLCKDs) are widely prescribed [8,9,10,11,12]. In particular Schiavo et al., have shown that a 4-week preoperative ketogenic diet is safe and effective at reducing body weight (−10.3%, *p* < 0.001, in males; −8.2%, *p* < 0.001, in females) and the left hepatic lobe volume (−19.8%, *p* < 0.001) in patients with obesity scheduled for BS [13]. Furthermore, Albanese et al., aiming to compare surgical outcome and weight loss in two groups of patients who were offered two different pre-operative kinds of diet (VLCD and VLCKD), reported that VLCKDs showed better results than VLCDs on surgical outcome, influencing the drainage output, post-operative hemoglobin levels, and hospital stay [14].

Evidence suggests that VLCKDs can be effective tools for positively managing weight loss, glycemic control, and lipid profile changes [15,16]. However, these beneficial effects can be limited by poor dietary adherence. In particular, cultural, religious, and economic barriers pose unique challenges to achieving nutritional compliance with VLCKDs [15,17]. A potential solution is represented by the enteral nutrition strategies.

Weight loss-based enteral nutrition strategies have been used in the treatment of obesity, showing promising results. In particular, Sukkar et al., assessing the feasibility of a protein-sparing modified diet delivered by naso-gastric tube enterally (with continuous feeding) in obesity treatment, showed that 10 days of enteral nutrition treatment followed by 20 days of a low-calorie diet was safe and effective at reducing total body weight and abdominal circumference, and in ameliorating the patients’ respiratory capacity without major complications and side effects [18].

Similarly, Castaldo et al. evaluated the effects of a carbohydrate-free diet delivered through enteral nutrition for t2 weeks, followed by an almost equivalent oral diet administered for a further 2 weeks in 112 patients, and reported a significant reduction in BMI and waist circumference with the amelioration of blood pressure values and insulin resistance without major complications [19].

Therefore, the enteral nutrition strategies could represent a possible alternative to other methodologies, in particular, when it is recommended to improve the patient’s adherence in following the prescribed diet before BS. However, to the best of our knowledge, there are no data concerning the use of enteral feeding approach in patients with obesity candidate to BS, neither on the dietary protocols to administer (e.g., hypocaloric or ketogenic), nor on how long to administer it before BS. Therefore, the aim of this study was to assess the clinical and metabolic impact, the efficacy, and the safety of ketogenic nutrition enteral protein (NEP) vs. nutritional enteral hypocaloric (NEI) protocols on patients with obesity candidate to BS.

## 2. Materials and Methods

### 2.1. Study Design and Characteristics of the Study Patients at Baseline

The study was conducted at Azienda Ospedaliera “San Giuseppe Moscati”, Avellino, Italy, between 1 October 2016 and 1 October 2019. Consecutive participants were recruited from the Division of Dietetics and Clinical Nutrition and the Division of General Surgery. All patients fulfilled the criteria declared by the International Federation for Surgery of Obesity for surgical treatment for morbid obesity [20,21].

In total, 62 patients were screened, while 60 patients were recruited and completed the intervention study. The inclusion criteria were: patient scheduled for BS after multi-disciplinary pre-operative evaluation, availability to long-term post-operative follow-up, normal kidney function serum creatinine ≤1.2 mg/dL and glomerular filtration rate ≥90 mL/min, and normal liver function (aspartate amino-transferase and/or alanine amino-transferase and/or gamma glutamyl transferase <2 × N). The exclusion criteria were: serum creatinine >1.2 mg/dL, liver failure (Child-Pugh ≥ A), insulin-dependent diabetes mellitus, atrioventricular block with QT > 0.44 ms, cardiac arrhythmias, moderate-severe cardiac failure, hypokalemia, chronic diarrhea or vomitus, 12-month previous cardio-vascular disease, pregnancy and/or lactation, current/previous neoplastic disease, psychiatric disorders, gastro-intestinal diseases, moderate-severe hypo-albuminemia (<3.0 mg/dL), 6 month previous diet-induced weight loss, and intragastric balloon.

The institutional ethics committee of Azienda Ospedaliera “San Giuseppe Moscati”, Avellino, Italy, approved the study protocol, which followed the Declaration of Helsinki, according to the International Guidelines of Good Clinical Practice and the regulations of clinical trials. Informed written consent was obtained from participants, after providing information about the nature, purpose, and procedures of the study (ClinicalTrials.gov Identifier: NCT02418975, 21 March 2017); Ethics Committee Approval CECN/132).

The patients were randomized 1:1 in 2 groups to undergo NEP (*n* = 31) or NEI treatments (*n* = 29). The naso-gastric tube (an 8-French polyurethane nasogastric tube) was placed after an overnight fast according to best clinical practice in day-hospital procedure. During the first visit, all patients were educated about the pump use, its feeding control, and any potential side effects (vomitus, nausea, etc.), receiving technical information for home use. Its use was necessary due to the need for precision of the daily calorie and lipid quotas to be administered.

### 2.2. Study Assessment and Endpoints

Assessments and measurements were performed at the baseline and after 4 weeks by the same nutritionist and radiologist in both groups. This study was blinded for the patient, surgical team, radiologist, and statistician.

The endpoints were to assess the clinical impact, efficacy, and safety of NEP vs. NEI protocols on patients with obesity candidate to BS. The duration of both the pre-operative nutritional interventions was 4 weeks. Body weight (BW), body mass index (BMI), waist circumference (WC), hip circumference (HC), and neck circumference (NC) were assessed at the baseline and at the 4-week follow-up. Furthermore, clinical parameters were assessed by blood tests, and patients were asked daily to report any side effects, using a self-administered questionnaire.

### 2.3. Safety

Patients were asked daily to report any side effects using a self-administered questionnaire in terms of asthenia, heartburn, nausea, vomiting, headache, dizziness, fainting, muscle cramps, hunger, orthostatic hypotension, palpitations, and constipation.

### 2.4. Dietary Interventions

NEP: Nutritional Enteral Protein (NEP) intervention consists of the continuous administration by a nasogastric probe, with the aid of a peristaltic feeding pump, of a highly hypocaloric glucidic liquid mixture (~5 kcal/kg/day) by enteral route, 2000 mL/per day (1.39 mL/min), based on 1.2 g protein/kg ideal body weight per day (calculated by Lorentz equation). The formula was made up of a fixed amount of some amino acids and a variable quantity of high-quality proteins (whey proteins). The other elements were coenzyme Q10, L-carnitine, α-linolenic acid, vitamin B6, and zinc. NEP was also accompanied by the daily oral administration of a nutritional supplement (FOS, 5000 mg; calcium carbonate, 1500 mg; magnesium carbonate, 850 mg; potassium bicarbonate, 500 mg; bicarbonate sodium, 1500 mg; potassium citrate, 500 mg; vitamin C, 180 mg; vitamin E, 30 mg; selenium, 55 mg; molybdenum, 50 µg; manganese, 1 mg; vitamin D3, 5 µg; and vitamin A, 800 µg) containing: alkalizing salts to increase the reserves of buffer substances in the body, minerals to supply the nutritional essential elements for maintaining a perfect mineral reserve, vitamins and trace elements with antioxidant activity, and intestinal transit regulatory fibers with prebiotic activity in order to promote the development of a healthy bacterial flora growth. In addition to these components, alga wakame and some herbal extracts were employed (horsetail, nettle, hawthorn, orthosiphon, and thistle) for draining, diuretic, and detoxifying actions. All patients orally took a gastric protector (proton pump inhibitor) and ursodeoxycholic acid (900 mg per day and 450 mg per day for those with and without documented liver disease). The duration of the pre-operative nutritional intervention was 4 weeks. The patients had been trained to freely drink water or unsweetened beverages (not tea or coffee), with a recommended minimum intake of 2 L per day. In patients with a history of kidney stones, the recommended water amount was 3 L per day. At the beginning of the treatment, therapy with hypoglycemic and diuretic drugs had been suspended. Treatments with antihypertensive and lipid-lowering drugs remained unchanged. During the nutritional intervention, the use of purgatives was not allowed.

NEI: NEI treatment consists of the continuous administration by a nasogastric probe, with the aid of a portable nutritional pump, of a liquid mixture with a balanced composition of macronutrients, low calorie (~20 kcal/kg/day), and normoproteins, based on 1 g protein/kg ideal body weight per day, and supplied with whey proteins. The infusion rate was 2000 mL/per day (1.39 mL/min). The duration of the pre-operative nutritional intervention was 4 weeks. NEI was also supplemented by the daily oral administration of a multivitamin-multimineral complex. All patients orally took a gastric protector (proton pump inhibitor) and ursodeoxycholic acid (900 mg per day and 450 mg per day for those with and without documented liver disease). The patients had been trained to freely drink water or unsweetened beverages (not tea or coffee), with a recommended minimum intake of 2 L per day. In patients with a history of kidney stones, the recommended water amount was 3 L per day. At the beginning of treatment, therapy with hypoglycemic and diuretic drugs have been suspended. The treatments with antihypertensive and lipid-lowering drugs remained unchanged. During the nutritional intervention, the use of purgatives was not allowed.

### 2.5. Anthropometric Evaluation of the Study Population

All of the participants had their heights and body weights (BWs) measured by calibrated flat scales equipped with a telescopic vertical steel stadiometer (SECA 711, Hamburg, Germany). The body mass index (BMI) was calculated as the weight (kg) divided by the height squared (kg/m^2^). A flexible plastic tape was used to assess the waist, hip, and neck circumferences (WC, HC, and NC, respectively).

### 2.6. Blood Tests of the Study Population

Blood samples were analyzed in the clinical laboratory using automated analyzers and available commercial kits. The following blood tests were performed: hemoglobin, hematocrit, glycated hemoglobin, glycemia, azotemia, creatinine, uricemia, total cholesterol, high-density lipoprotein (HDL) cholesterol, low-density lipoprotein (LDL) cholesterol, triglycerides, apolipoproteins Apo A1 and Apo B, serum aspartate aminotransferase (AST), alanine aminotransferase (ALT), gamma glutamyl transferase (γGT), sideremia, total proteins, transferrin, albumin, sodium, potassium, calcium, magnesium, phosphorus, insulin, homeostasis model assessment insulin resistance (HOMA index), and ferritin.

### 2.7. Ultrasound Measurement

For the assessment of the aortomesenteric fat thickness (AMFT), steatosis grade, and left hepatic lobe volume, according to a previous method [22], ultrasound measurements were performed using an ultrasonographic system (Hitachi EUB-8500, Hitachi Medical Systems America, Inc., Twinsburg, OH, USA).

### 2.8. Statistical Analysis

Data were analyzed by the retrospective analysis of a prospective database. The statistical analysis, data visualization, and predictive analysis were performed using Statgraphics software (Statgraphics Technologies, Inc., The Plains, VA, USA). The characteristics of the population included in this study were analyzed using descriptive techniques. The results were expressed as the average mean and standard deviation. The statistical analysis of the parametric data was carried out with the Student’s *t* test, comparing the data at the baseline and after 4 weeks within the groups and between the NEP and NEI groups using the Mann-Whitney U test. The values of *p* < 0.05 were considered statistically significant, with the relative confidence interval at 95%. Furthermore, any *p*-value less than 0.001 was conventionally stated merely as *p* < 0.001.

## 3. Results

### 3.1. Impact of NEP and NEI on BW, BMI, WC, HC, and NC

As shown in Table 1, before surgery, the NEP and NEI groups were comparable in terms of age, BW, and BMI.

As shown in Table 2, compared to the baseline, the BW, BMI, WC, HC, and NC were significantly reduced in both groups studied (*p* < 0.001). However, we did not find any significative differences between the NEP and NEI groups in terms of weight loss (*p* = 0.559), BMI (*p*= 0.383), WC (*p* = 0.779), and HC (*p* = 0.559), while a statistically significant difference was found in terms of the NC (*p* = 0.011).

### 3.2. Impact of NEP and NEI on Patient’s Clinical Parameters and Safety

As reported in Table 3, we found a significant amelioration of the general clinical status in both groups studied. However, the NEP group showed a significant improvement in terms of glycemic and lipid profiles when compared with the NEI group.

In particular, as shown in Figure 1A–C, a statistically significant difference was found in terms of glycemia (NEP, −16% vs. NEI, −8.5%, *p* < 0.001), insulin (NEP, −49.6% vs. NEI, −17.8%, *p* < 0.0028), and the HOMA index (NEP, −57.7% vs. NEI, −24.9%, *p* < 0.001), respectively.

Furthermore, as shown in Figure 2A–D, a statistically significant difference was additionally found in terms of the total cholesterol (NEP, −24.3% vs. NEI, −2.8%, *p* < 0.001), low-density lipoprotein (NEP, −30.9% vs. NEI, 1.96%, *p* < 0.001), apolipoprotein A1 (NEP, −24.2% vs. NEI, −7%, *p* < 0.001), and apolipoprotein B (NEP, −23.1% vs. NEI, −2.3%, *p* < 0.001), respectively. Regarding safety, no important side effects were reported. The most frequent side effect was constipation 8%, followed by headache 7% and nausea 2%, especially during the first days.

### 3.3. Impact of NEP and NEI on AMFT, Steatosis Grade, and Left Lobe Liver Volume

As shown in Table 4, we did not find any significative differences between the NEP and NEI groups in terms of aortomesenteric fat thickness (*p* = 0.332), degree of steatosis (*p* = 0.616), and left hepatic lobe volume (*p* = 0.264).

## 4. Discussion

The present study indicates that enteral feeding is an effective and safe treatment before BS, with NEP leading to better clinical results than NEI on glycemic and lipid profiles. The role of dietary therapies before BS is widely acknowledged not only for achieving weight loss and body metrics amelioration, but also in reducing the risk of intra- and peri-operative complications, improving patients’ metabolic profiles, cardiovascular and respiratory conditions, and reducing the inflammatory status [3,6,8,9,17].

The safety and the efficacy of enteral feeding in patients with obesity has been seldom investigated [18,19]. However, to the best of our knowledge, there are no data concerning the use of enteral feeding approach in patients with obesity candidate to BS, neither on the dietary protocol to administer (e.g., hypocaloric, or ketogenic), nor on how long to administer it before BS.

The weight loss obtained in our study was similar to those reported in previous studies [10,23]. Furthermore, herein we found that, compared to the baseline, WC, HC, and NC were significantly reduced in both groups studied. However, we did not find any significative difference between the NEP and NEI groups in terms of weight loss, WC, and HC, while a statistically significant difference was found in terms of NC. The NC data are clinically significant in patients with obesity candidate to BS. In fact, today, most types of BS are performed laparoscopically. However, the key element in laparoscopic surgery is the creation of pneumoperitoneum and carbon dioxide is commonly used for insufflation. The various effects of induction of pneumoperitoneum can result in respiratory embarrassment and cardiovascular changes best managed by the use of general anesthesia with endotracheal intubation. As matter of fact, NC represents a predictor of difficult intubation and difficult mask ventilation in patients with morbid obesity [24].

Concerning patients’ clinical status, the NEP strategy showed a higher impact than NEI on several parameters, such as on glycemic and lipid profiles. This was in accordance with a recent meta-analysis of clinical trials conducted by Alarim et al. with the aim to look at the published literature and summarize the interventional trials that use the ketogenic diet for glycemic control and lipid profiles, concluding that the ketogenic diet is superior to other nutritional strategies in terms of glycemic control and lipid profile improvements [25]. It is well known that the ketogenic diet represents a nutritional strategy based on the reduction in dietary carbohydrates, which induces the body to produce the glucose necessary for survival and to increase the energy consumption of fats contained in adipose tissue. Therefore, in agreement with the literature, the significant amelioration in both the glycemic and lipid profiles is, at least in part, due to the reduction in carbohydrate intake, leading to reduced blood glucose and shifting the basic metabolism of energy from glucose to ketone bodies. Furthermore, in accordance with our data, this decrease in blood glucose leads to improved insulin resistance as well [26].

From a surgical point-of-view, liver steatosis in patients suffering from morbid obesity undergoing BS increases the liver volume and may complicate the surgical procedure when the liver’s left lateral section is massively enlarged, limiting the access to the esophagogastric junction and increasing the risk of laceration of the soft fatty liver with consequent bleeding [6,7]. In turn, these difficulties may result in an increased operative time, suboptimal surgery, and an increased rate of conversion to open surgery [8]. The present study indicates that both the NEP and NEI approach were effective at reducing left hepatic lobe volume, steatosis grade, and AMFT in patients with obesity scheduled for BS, with the NEP intervention allowing for a higher reduction in the left hepatic lobe volume (−31.4 vs. −18.5%, respectively), steatosis grade (−20.8 vs. −16.1%, respectively), and AMFT (−28.9 vs. −17.3%, respectively) than NEI. This has a huge clinical value as AMFT values represent an important component and cause of metabolic syndrome and are associated with greater cardiometabolic risk [22].

In accordance with the studies of Castaldo et al. [27,28], in terms of patients’ adherence, NEP and NEI interventions were safe, feasible, and well-tolerated. Only one patient discontinued the study (NEI group). Therapeutic adherence includes patient adherence not only with respect to medication, but also regarding diet, exercise, or lifestyle changes. Thus, therapeutic nonadherence occurs when an individual’s health-seeking or maintenance behavior lacks congruence with the recommendations prescribed by a healthcare provider [29]. Herein, considering that both the NEP and NEI treatments were performed using the nasogastric tube technique in the hospitalization regimen, we did not need to indirectly measure the patients’ adherence by questionnaires. Regarding safety, no important side effects were reported. The main strengths of the present study are:

1.The enteral nutrition strategies could represent a possible alternative to other methodologies, in particular when it is recommended to improve the patient’s adherence to following the prescribed diet before BS.2.Enteral feeding is an effective and safe treatment before BS, with NEP warranting better clinical results than NEI on glycemic and lipid profiles.3.Regarding safety, no important side effects were reported.

This study has some limitations, including the small number of patients studied and the short-term follow-up that did not allow us to draw definitive conclusions. Furthermore, we did not include the total lymphocyte count nor the serum prealbumin concentration as laboratory markers of the patients‘ nutrition. However, concerning the total lymphocyte count, as suggested in the recent literature, it did not represent a specific and insensitive marker of the nutritional status [30]. Furthermore, as described in the American Society for Parenteral and Enteral Nutrition (ASPEN) position paper, despite serum albumin and prealbumin, well-known visceral proteins, being traditionally considered as useful biochemical laboratory values in a nutrition assessment, the recent literature disputes this contention. In particular, the ASPEN position paper clarifies that these proteins characterize inflammation rather than describe nutrition status or protein-energy malnutrition. Obesity is characterized by chronic low-grade inflammation and, as such, hepatic reprioritization of protein synthesis occurs, resulting in lower serum concentrations of albumin and prealbumin. In addition, the redistribution of serum proteins occurs because of an increase in capillary permeability. There is an association between inflammation and malnutrition, but not between malnutrition and visceral-protein levels. These proteins correlate well with patients’ risk for adverse outcomes rather than with protein-energy malnutrition. Therefore, serum albumin and prealbumin should not serve as proxy measures of the total body protein or total muscle mass and should not be used as nutrition markers [31,32].

## 5. Conclusions

Ketogenic diet-induced-weight loss before BS has beneficial effects on the reduction in liver volume, metabolic profile, and intra- and post-operative complications. However, these beneficial effects can be limited by poor dietary adherence. A potential solution in patients showing a poor adherence to following the prescribed diet could be represented by the enteral nutrition strategies.

Based on our findings, despite the small sample size, we were able to support the hypothesis that enteral feeding is an effective and safe treatment before BS, with NEP achieving better clinical results than NEI on glycemic and lipid profiles. Further and larger randomized clinical trials are needed to confirm these preliminary data.

## Figures and Tables

**Figure 1 nutrients-15-01492-f001:**
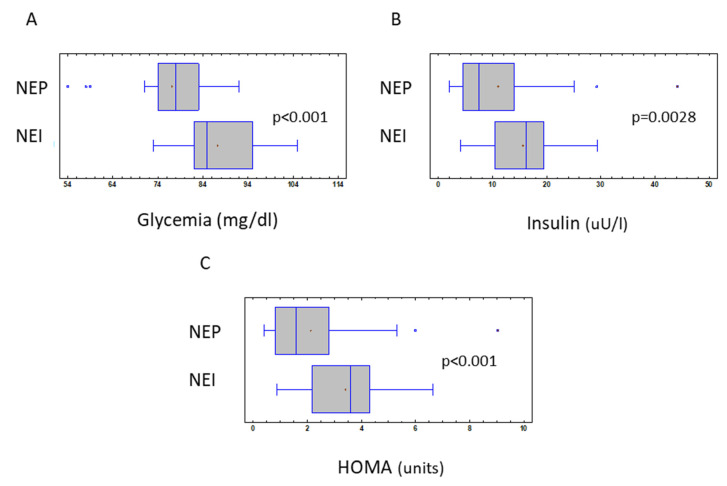
Box plots showing baseline and 4-week follow-up changes in glycemia (**A**), insulin (**B**), and HOMA index (**C**) in both groups studied. A box extending from the lower quartile to the upper quartile. The middle 50% of the data values are, thus, covered by the box, with a vertical line at the location of the sample median, which divides the data in half. NEP = nutritional enteral protein; NEI = nutritional enteral hypocaloric.

**Figure 2 nutrients-15-01492-f002:**
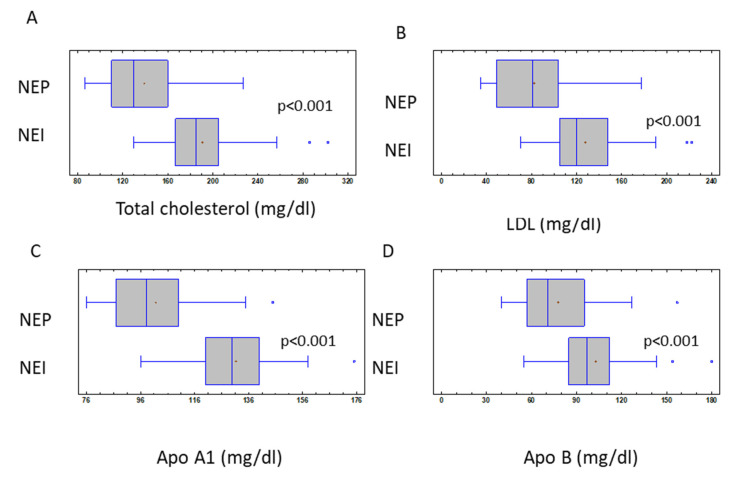
Box plots showing baseline and 4-week follow-up changes in total cholesterol (**A**), LDL (**B**), Apo A1 (**C**), and Apo B (**D**) in both groups studied. A box extending from the lower quartile to the upper quartile. The middle 50% of the data values are, thus, covered by the box, with the vertical line at the location of the sample median, which divides the data in half. NEP = nutritional enteral protein, NEI = nutritional enteral hypocaloric.

**Table 1 nutrients-15-01492-t001:** Characteristics of participants.

	NEP (*n* = 31)	NEI (*n* = 29)	*p* (NEP vs. NEI)
Sex (male/female, *n*)	6/25	6/23	/
Age (mean ± SD, years)	36.33 ± 10.20	39.31 ± 6.91	0.8013
Body weight (mean ± SD, kg)	130.47 ± 23.10	124.09 ± 17.20	0.5590
BMI (mean ± SD, kg/m^2^)	47.89 ± 6.99	45.19 ± 4.87	0.3827

BMI = body mass index, SD = standard deviation.

**Table 2 nutrients-15-01492-t002:** Anthropometric evaluation of the study population.

Clinical Parameters	Groups	Baseline	Follow-Up 4 Weeks	*p* (Baseline vs. Follow-Up)	*p* (NEP vs. NEI)
BW, Kg	NEP NEI	130.47 ± 23.10 124.09 ± 17.20	119.70 ± 21.71 117.23 ± 16.57	<0.001 * <0.001 *	0.559
BMI, Kg/m^2^	NEP NEI	47.89 ± 6.99 45.19 ± 4.87	43.95 ± 6.57 42.68 ± 4.58	<0.001 * <0.001 *	0.383
WC, cm	NEP NEI	137.52 ± 14.47 132.14 ± 11.02	127.45 ± 14.09 126.60 ± 10.99	<0.001 * <0.001 *	0.779
HC, cm	NEP NEI	141.31 ± 14.95 136.62 ± 13.00	133.58 ± 14.29 131.07 ± 12.31	<0.001 * <0.001 *	0.559
NC, cm	NEP NEI	41.53 ± 4.72 42.33 ± 3.19	38.58 ± 4.10 40.62 ± 3.11	<0.001 * <0.001 *	0.011 *

* *p*-value < 0.05.

**Table 3 nutrients-15-01492-t003:** Blood tests of the study population.

Clinical Parameters	Groups	Baseline	Follow-Up 4 Weeks	*p* (Baseline vs. Follow-Up)	*p* (NEP vs. NEI)
Hemoglobin, g/dL	NEP NEI	13.94 ± 1.66 14.53 ± 4.40	13.71 ± 1.49 13.73 ± 1.34	0.1338 0.3283	0.8940
Hematocrit, %	NEP NEI	40.85 ± 4.13 40.86 ± 3.95	40.63 ± 3.81 41.23 ± 3.67	0.6128 0.1987	0.6895
Glycated Hemoglobin, %	NEP NEI	6.02 ± 0.92 5.73 ± 0.87	5.69 ± 0.64 5.51 ± 0.55	<0.001 * 0.0104 *	0.2686
Glycemia, mg/dL	NEP NEI	91.81 ± 25.14 95.38 ± 33.77	77.10 ± 9.31 87.24 ± 8.23	0.0012 * 0.2380	<0.001 *
Insulin, µU/L	NEP NEI	21.91 ± 16.66 19.07 ± 7.16	11.05 ± 9.60 15.68 ± 6.26	<0.001 * 0.0172 *	0.0028 *
HOMA Index	NEP NEI	5.03 ± 4.53 4.54 ± 2.27	2.12 ± 1.94 3.41 ± 1.47	<0.001 * 0.0126 *	<0.001 *
Azotemia, mg/dL	NEP NEI	28.55 ± 6.19 30.31 ± 5.95	25.29 ± 6.64 27.41 ± 8.13	0.0611 0.0246 *	0.2105
Creatinine, mg/dL	NEP NEI	0.69 ± 0.12 0.72 ± 0.12	0.67 ± 0.11 0.80 ± 0.22	0.1674 0.0267 *	0.0076 *
Uricemia, mg/dL	NEP NEI	5.11 ± 1.21 5.23 ± 1.20	6.24 ± 2.69 5.67 ± 1.24	0.0113 * 0.0134 *	0.9233
Total Cholesterol, mg/dL	NEP NEI	184.00 ± 34.74 196.07 ± 32.57	139.32 ± 38.14 190.62 ± 40.06	<0.001 * 0.3758	<0.001 *
LDL, mg/dL	NEP NEI	119.23 ± 38.46 130.31 ± 31.35	82.39 ± 35.76 127.76 ± 38.31	<0.001 * 0.2567	<0.001 *
HDL, mg/dL	NEP NEI	49.26 ± 12.56 50.52 ± 12.23	39.10 ± 11.70 47.45 ± 11.84	<0.001 * 0.0117 *	0.0107 *
Triglycerides, mg/dL	NEP NEI	120.10 ± 55.60 124.21 ± 70.15	100.68 ± 52.85 105.28 ± 44.88	0.0200 * 0.0920	0.5343
Apo A1, mg/dL	NEP NEI	133.97 ± 21.19 141.28 ± 18.44	101.58 ± 17.62 131.31 ± 17.80	<0.001 * <0.001 *	<0.001 *
Apo B, mg/dL	NEP NEI	100.94 ± 28.51 105.38 ± 26.60	77.65 ± 27.04 102.97 ± 26.74	<0.001 * 0.5500	<0.001 *
AST, U/L	NEP NEI	20.45 ± 9.55 20.17 ± 9.45	25.35 ± 9.08 21.52 ± 7.19	0.0095 * 0.2999	0.0818
ALT, U/L	NEP NEI	28.48 ± 18.94 25.28 ± 14.20	35.00 ± 27.55 26.17 ± 11.19	0.1567 0.7116	0.4774
γGT, U/L	NEP NEI	25.55 ± 16.36 29.03 ± 10.07	16.19 ± 9.08 22.31 ± 12.14	<0.001 * 0.0882	0.3420
Sideremia, mcg/dL	NEP NEI	67.71 ± 26.34 69.28 ± 19.60	52.74 ± 17.95 79.24 ± 27.42	0.0041 * 0.0123 *	<0.001 *
Total Proteins, g/dL	NEP NEI	7.18 ± 0.35 7.08 ± 0.35	6.97 ± 0.86 7.17 ± 0.33	0.1440 0.1068	0.3936
Transferrin, mg/dL	NEP NEI	279.39 ± 45.87 274.79 ± 42.29	245.39 ± 53.56 261.25 ± 63.79	<0.001 * 0.2117	0.0711
Albumin, g/dL	NEP NEI	4.15 ± 0.24 4.12 ± 0.32	4.21 ± 0.33 4.24 ± 0.32	0.3912 0.0108 *	0.8590
Sodium, mmol/L	NEP NEI	138.52 ± 2.35 139.21 ± 1.59	138.06 ± 2.61 139.34 ± 2.00	0.3602 0.6626	0.0197 *
Potassium, mequ/L	NEP NEI	4.54 ± 0.30 4.53 ± 0.30	4.55 ± 0.34 4.46 ± 0.31	0.7985 0.3108	0.0935
Calcium, mg/dL	NEP NEI	9.31 ± 0.45 9.31 ± 0.30	9.38 ± 0.46 9.46 ± 0.34	0.3790 0.0183 *	0.7608
Magnesium, mg/dL	NEP NEI	2.07 ± 0.08 2.07 ± 0.15	2.02 ± 0.16 2.10 ± 0.13	0.1562 0.2171	0.0789
Phosphorus, mg/dL	NEP NEI	3.25 ± 0.56 3.38 ± 0.38	3.53 ± 0.51 3.35 ± 0.43	0.0155 * 0.6333	0.1562
Ferritin, ng/mL	NEP NEI	151.94 ± 66.79 91.97 ± 30.38	151.74 ± 69.60 104.07 ± 38.72	0.9798 0.0614	0.0019 *

* *p*-value < 0.05.

**Table 4 nutrients-15-01492-t004:** Liver ultrasound measurements.

Clinical Parameters	Groups	Baseline	Follow-Up 4 Weeks	*p* (Baseline vs. Follow-Up)	*p* (NEP vs. NEI)
AMFT, mm	NEP NEI	22.32 ± 11.64 21.43 ± 7.72	15.86 ± 5.12 17.70 ± 6.87	0.0015 * <0.001 *	0.3319
Steatosis, grade	NEP NEI	2.40 ± 0.42 2.36 ± 0.35	1.90 ± 0.45 1.98 ± 0.39	<0.001 * <0.001 *	0.6156
Left hepatic lobe volume, cm^3^	NEP NEI	407.39 ± 125.77 385.14 ± 123.90	279.58 ± 99.68 313.72 ± 113.98	<0.001 * <0.001 *	0.2640

* *p*-value < 0.05.

## Data Availability

The data included in this manuscript were derived from the University database. We are not authorized to share the data with third party organizations. However, the corresponding author is available to provide any explanation to the editor if requested.

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
