# Peer review of "Clinical Impact of Enteral Protein Nutritional Therapy on Patients with Obesity Scheduled for Bariatric Surgery: A Focus on Safety, Efficacy, and Pathophysiological Changes"

_nutrients, 2023, doi:10.3390/nu15061492_

Round 1

Reviewer 1 Report

This is a very interesting original prospective randomized 1:1 study regarding the assessment of the clinical impact, efficacy, and safety of ketogenic nutrition enteral protein (NEP) vs nutritional enteral hypocaloric (NEI) protocols on patients with obesity candidate to BS.

The following issues should be adressed:

  1. The analyzed clinical, radiological and laboratory parameters were assessed at baseline and after 4-weeks follow-up. It does mean that the duration of nutritional intervention was 4 weeks? Information on the duration of nutritional intervention should be added in the paragraph 2.2. Study Assessment and Endpoints.

  2. There are a lot laboratory parameters assessed in this study. Have you assessed some other nutritional parameters such as total lymphocyte count in the peripheal blood and serum prealbumin concentration? They should be added and compared between two groups.

  3. Information regarding the immune status including baseline and after 4-weeks follow-up neutrophil/lymphocyte ratio (NLR), platelet/lymphocyte ratio (PLR), and monocyte/lymphocyte ratio (MLR) ratios would be interesting.

  4. More information regarding limitations as well as strengths of this study should be presented in the discussion.

Author Response

Dear Reviewer

We sincerely thank you for your review and assessment of our manuscript. We have revised our manuscript accordingly and we hope you find these revisions suitable. In the revised manuscript, we carefully addressed the issues raised and we have tried to do our best to provide a point-by-point reply.

Reviewer 2 Report

Thank you for submitting the manuscript "Clinical Impact of Enteral Protein Nutritional Therapy-Induced Weight Loss on Patients with Obesity Scheduled for Bariatric Surgery: A Focus on Safety, Efficacy, and Pathophysiological Changes" to Nutrients.

Although the research is very interesting and the data are probably unpublished, the work lacked a group without the use of enteral diet so that it could be really affirmed that the use of naso-gastric tube can result in better effects in the loss of mass before AB.

It would be interesting if the authors better justified the use of extra procedures with the patient who will already be submitted to AB. It would be easier to check by numbers: what percentage of patients have improved conditions with the use of enteral nutrition? Weight loss will already happen with the low-calorie diet, it is not a reason to perform invasive procedures with the patient.

Line#34: if there was no significant difference for weight loss, why is this parameter included in the title of the manuscript? My suggestion is to change the title to one that really describes some important finding of this work.

Line#43: Please use parentheses for the p-value.

Line#48: Avoid keywords that are already in the title.

Line#53: indicate the desired percentage of preoperative weight loss.

Line#63: two that

Line#64: What is this safe reported by the authors? This is also in the title, however it does not seem that this safety was evaluated in the present work.

Line#68: parentheses.

Line#148: include diet infusion rate.

Author Response

(The authors gave the same response as above.)

Round 2

Reviewer 1 Report

The authors have improved their manuscript according to some my suggestions, but there is no explanation point by point of the revisions to the manuscript and authors' responses to my comments. It should be added in a cover letter.